# Slack-Free Spiking Neural Network Formulation for Hypergraph Minimum Vertex Cover

**Tam Ngoc-Bang Nguyen[1], Anh-Dzung Doan[1], Zhipeng Cai[2], Tat-Jun Chin[1]**

[1] Australian Institute for Machine Learning, The University of Adelaide,

[2] Intel Labs

{tam.nb.nguyen,dzung.doan,tat-jun.chin}@adelaide.edu.au

zhipeng.cai@intel.com

## Abstract

Neuromorphic computers open up the potential of energy-efficient computation using spiking neural networks (SNN), which consist of neurons that exchange spike-based information asynchronously. In particular, SNNs have shown promise in solving combinatorial optimization. Underpinning the SNN methods is the concept of energy minimization of an Ising model, which is closely related to quadratic unconstrained binary optimization (QUBO). Thus, the starting point for many SNN methods is reformulating the target problem as QUBO, then executing an SNN-based QUBO solver. For many combinatorial problems, the reformulation entails introducing penalty terms, potentially with slack variables, that implement feasibility constraints in the QUBO objective. For more complex problems such as hypergraph minimum vertex cover (HMVC), numerous slack variables are introduced which drastically increase the search domain and reduce the effectiveness of the SNN solver. In this paper, we propose a novel SNN formulation for HMVC. Rather than using penalty terms with slack variables, our SNN architecture introduces additional spiking neurons with a constraint checking and correction mechanism that encourages convergence to feasible solutions. In effect, our method obviates the need for reformulating HMVC as QUBO. Experiments on neuromorphic hardware show that our method consistently yielded high quality solutions for HMVC on real and synthetic instances where the SNN-based QUBO solver often failed, while consuming measurably less energy than global solvers on CPU.

## 1   Introduction

Neuromorphic computing research aims to develop computational models inspired by neural architectures found in nature. Spiking neural networks (SNN), in which a network of processing units (neurons) asynchronously transmit spike-based messages to each other [34], is a notable representative of the neuromorphic paradigm. Through parallelism, stochastic behavior, event-driven computing and other biologically-inspired properties, SNNs promise higher energy efficiency than conventional computing, which includes the successful artificial neural networks (ANN) [16].

The advent of neuromorphic computing hardware that can implement SNNs has boosted research in the area. Notable examples include IBM TrueNorth [25] and Intel Loihi [12, 30, 35]. Rigorous benchmarking [13] indicate the promise of SNNs in delivering energy efficient computations, which not only benefit edge computing applications, but also reducing the energy consumption of data centers. The recent introduction of Intel Hala Point [1], the world's largest neuromorphic supercomputer with 1.15 billion neurons and 138.2 billion synapses, is a testament of confidence in the technology.

Two major classes of problems have been explored for SNNs: machine learning and combinatorial optimization [13]. Representative approaches in the former include training SNNs using asynchronous

38th Conference on Neural Information Processing Systems (NeurIPS 2024).

variants of backpropagation, and converting pre-trained ANNs into equivalent SNNs for deployment on neuromorphic hardware. Works in the latter develop SNNs to solve specific optimization problems, where the spike-based temporal information processing is exploited to achieve computational benefits over classical methods. Our work focuses on combinatorial optimization.

The concept of energy minimization of the Ising model underpins SNN techniques for combinatorial optimization. The task is closely related to quadratic unconstrained binary optimization (QUBO)

$$\min_{\mathbf{z} \in \{0,1\}^N} \mathbf{z}^T \mathbf{Q} \mathbf{z}, \tag{1}$$

where $\mathbf{Q} \in \mathbb{Z}^{N \times N}$ is a symmetric coefficient matrix (we restrict $\mathbf{Q}$ to integers to facilitate the SNN treatment; integers are sufficient nonetheless for the targeted combinatorial problem). Several SNNs have been developed to solve QUBO [6, 14, 3, 32]; Sec. 3.2 will describe such a method. To bring such SNNs to bear on other combinatorial problems, the problems must first be reformulated into QUBO [11, 22]. For problems with feasibility constraints, penalty terms and often slack variables will be added to the QUBO objective to implement the constraints. Examples are provided below.

**Problem 1** (Minimum vertex cover (MVC)). Let $G = (V, E)$ be a graph with vertices $V = \{1, \ldots, N\}$ and edges $E = \{e_1, \ldots, e_K\}$, where each $e_k = \langle i^{(k)}, j^{(k)} \rangle \subset V$ connects two vertices. A vertex cover of $G$ is a subset $Z$ of $V$ such that every edge is incident with at least a vertex in $Z$. The goal of MVC is to find the vertex cover with the smallest number of vertices. Let

$$\mathbf{z} = [z_1, \ldots, z_N]^T \in \{0,1\}^N \tag{2}$$

be a binary vector whose role is to select a subset of $V$, where $z_i = 1$ indicates that the $i$-th vertex is selected, and $z_i = 0$ means otherwise. MVC can be expressed as

$$\begin{aligned} \min_{\mathbf{z} \in \{0,1\}^N} \quad & \|\mathbf{z}\|_1 \\ \text{s.t.} \quad & (1 - z_{i(k)})(1 - z_{j(k)}) = 0, \quad \forall k = 1, \ldots, K. \end{aligned} \tag{3}$$

Note that $\|\mathbf{z}\|_1 = \|\mathbf{z}\|_2$ for binary $\mathbf{z}$ and the equality constraints in (3) are quadratic in $\mathbf{z}$. Rewriting the equality constraints as penalty terms, we obtain the QUBO

$$\min_{\mathbf{z} \in \{0,1\}^N} \quad \|\mathbf{z}\|_2^2 + \lambda \sum_{k=1}^{K} (1 - z_{i(k)})(1 - z_{j(k)}). \tag{4}$$

Intuitively, solutions $\mathbf{z}$ that violate the constraints in (3) will raise the total cost in (4) and hence be penalized. The quantity $\lambda \in \mathbb{Z}$ is the penalty weight that controls the degree of penalization due to constraint violations. Note that while (4) is not exactly in the form (1) due to the existence of a constant, the remaining steps to rearrange (4) to (1) are minor; see supplementary material. □

**Problem 2** (Hypergraph minimum vertex cover (HMVC)). Let $H = (V, F)$ be an $r-$uniform hypergraph with vertices $V = \{1, \ldots, N\}$ and hyperedges $F = \{f_1, \ldots, f_K\}$, where each $f_k \subset V$ is incident with exactly $r$ vertices, $r \geq 2$. A vertex cover of $H$ is a subset $Z$ of $V$ such that every hyperedge is incident with at least a vertex in $Z$. The goal of HMVC is to find the vertex cover with the smallest number of vertices. Note that HMVC reduces to MVC if $r = 2$. For each hyperedge $f_k$, define a binary vector

$$\mathbf{b}^{(k)} = [b_1^{(k)}, \ldots, b_N^{(k)}]^T \in \{0,1\}^N, \tag{5}$$

where $\|\mathbf{b}^{(k)}\|_1 = r$, and $b_i^{(k)} = 1$ means that vertex $i$ is incident to $f_k$ and $b_i^{(k)} = 0$ means otherwise. HMVC can then be written as

$$\begin{aligned} \min_{\mathbf{z} \in \{0,1\}^N} \quad & \|\mathbf{z}\|_1 \\ \text{s.t.} \quad & {\mathbf{b}^{(k)}}^T \mathbf{z} \geq 1, \quad \forall k = 1, \ldots, K, \end{aligned} \tag{6}$$

which is a 0-1 integer linear program (ILP). Since $r$ variables ($r \geq 2$) exist in each linear inequality constraint, in general it is not possible to express it as a quadratic equality constraint, *cf*. (3). Instead, the path to QUBO will involve converting each inequality into an equality constraint

$$ {\mathbf{b}^{(k)}}^T \mathbf{z} - \mathbf{1}_{r'}^T \mathbf{y}^{(k)} = 1, \tag{7}$$

which requires introducing $r'$ binary slack variables for each $f_k$

$$\mathbf{y}^{(k)} = [y_1^{(k)}, \ldots, y_{r'}^{(k)}]^T \in \{0,1\}^{r'}, \tag{8}$$

where $r' = r - 1$ and $\mathbf{1}_{r'}$ is a column vector of 1 of size $r'$. Installing the equality constraints as quadratic penalty terms, we obtain the QUBO

$$\min_{\mathbf{z} \in \{0,1\}^N, \{\mathbf{y}^{(k)}\}_{k=1}^K \in \{0,1\}^{r' \times K}} \|\mathbf{z}\|_2^2 + \lambda \sum_{k=1}^K \left( \mathbf{b}^{(k)T} \mathbf{z} - \mathbf{1}_{r'}^T \mathbf{y}^{(k)} - 1 \right)^2, \tag{9}$$

where $\lambda \in \mathbb{Z}$ is the penalty weight. See supp. material for rewriting the QUBO in form (1). $\qquad\square$

A major difference between the QUBO reformulations for MVC and HMVC is that the latter requires slack variables, the quantity of which scales linearly with the number $K$ and degree $r$ of hyperedges. This significantly increases the search space and difficulty of the optimization. As we will show in Sec. 5, the existing SNN-based QUBO solver [3] is unable to satisfactorily solve HMVC based on (9). This presents an obstacle towards applying SNNs to a combinatorial problem with many practical applications, *e.g.*, computational biology [9], computer network security [19], resource allocation [7] and social network analysis [23]. Fundamentally, HMVC is a general optimization problem that encompasses several related formulations, such as set cover, hitting set, and traversal [5], giving it wide applicability. The issue calls for a more effective SNN for HMVC.

## 1.1 Contributions

To solve HMVC more effectively, we propose an SNN that is composed of a mix of non-equilibrium Boltzmann machine (NEBM) spiking neurons [32] and a custom *feedback* spiking neuron. One feedback neuron is introduced for each constraint in (6) to check for constraint violations and encourage the overall state to return to feasibility. A major benefit of our handcrafted SNN is obviating the need to reformulate HMVC as QUBO based on the penalty method, which not only avoids the usage of slack variables, but also removes the necessity to tune the penalty weight.

Results on Intel Loihi 2 [30] indicate that our SNN significantly outperformed the QUBO approach on HMVC, in that our method consistently yielded high-quality solutions on synthetic and real problem instances where the SNN-based QUBO solver [3] often failed to return feasible results. Moreover, our SNN consumed measurably much less energy than global solvers on CPU.

Our work follows the spirit of other handcrafted SNNs for combinatorial optimization [8, 31, 17, 21, 36]. However, such works have mainly been targeted at constraint satisfaction problems, whereas our SNN is aimed at a graph-based optimization problem, *i.e.*, HMVC (more details in Sec. 2).

## 2 Related work

Previous studies have shown that an SNN with a topology corresponding to the matrix $\mathbf{Q}$ can efficiently solve QUBO [3, 6, 14]. This enables neuromorphic computing to solve combinatorial problems that can be encoded as QUBO, such as graph partitioning [26]. In addition, via the penalty method [29, Chap. 17], other combinatorial problems with constraints can be reformulated into QUBO [15]. Problem 1 has discussed this for MVC, which has been evaluated on a neuromorphic processor [11]. Other works that employed QUBO reformulation include [32, 22] who solved maximum independent set and ILP. However, more complex problems will require the introduction of slack variables, which increases the search space and difficulty for an SNN solution. Problem 2 has illustrated this for HMVC.

The majority of handcrafted SNNs for combinatorial optimization aim to solve constraint satisfaction problems (CSPs). Jonke *et al.* [21] proposed a stochastic SNN for traveling salesman problem. Since then, several SNN solvers have been proposed for CSPs, such as, Sudoku [8, 31], graph coloring [17], and Boolean satisfiability problem [21, 36]. These existing approaches share a common strategy: constructing a specific SNN topology that is strongly tailored to the constraints of each CSP. This is to ensure that these SNN solvers can seek valid values for a set of variables that satisfy the specified constraints. In other words, the primary objective of these handcrafted SNNs is to find feasible solutions to the combinatorial optimization problem. Although our approach shares a similar spirit

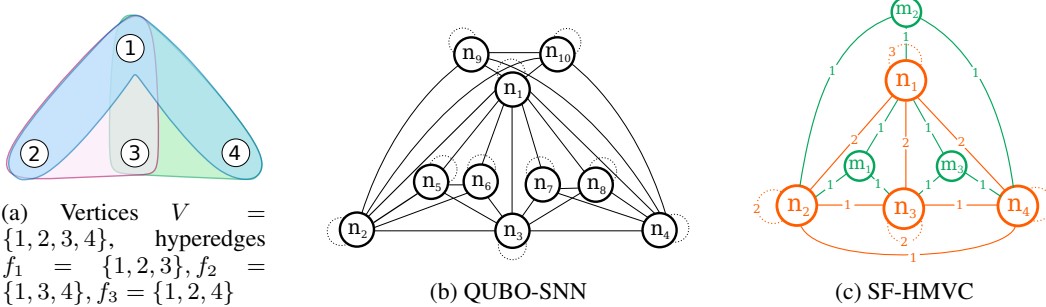

(a) Vertices $V = \{1, 2, 3, 4\}$, hyperedges $f_1 = \{1, 2, 3\}, f_2 = \{1, 3, 4\}, f_3 = \{1, 2, 4\}$

(b) QUBO-SNN

(c) SF-HMVC

Figure 1: (a) HMVC input hypergraph: a 3-uniform hypergraph with 4 vertices and 3 hyperedges. (b) SNN for QUBO (9) derived from (a). Both the variables $z_i$ and introduced slack variables are NEBM neurons. The edge weights are omitted for brevity. (c) The SNN from our method for input (a). NEBM and FB neurons are in orange and green resp. Dashed circles indicate self-connections.

to these handcrafted SNNs in that we directly construct SNN topology using constraints, our SNN topological graph is specifically designed to seek not only a feasible solution but also the optimal one.

The QUBO formulation is amenable to Ising-specific analog and digital hardware solvers, including spintronics, memristors, and quantum annealers [27]. However, given our focus on SNNs, we leverage the state-of-the-art Loihi 2 neuromorphic accelerator to implement the SNN solutions for both QUBO and our method, due to its high flexibility in SNN design and programmable neuron model. For a comprehensive overview of Loihi 2's capabilities, we refer readers to [24, 32, 35].

There is a large body of work on SNNs for machine learning; we refer to [13] for a thorough survey. While SNNs are usually less accurate than ANNs [28], under certain temporal dynamics, SNNs can surpass ANNs. SNNs have also been used for robotics. For instance, [24] implemented their SNN-based quadratic programming solver for model predictive control on Loihi 2, achieving two orders-of-magnitude gains in energy-delay product compared to CPU solvers. Another study demonstrated the applications of SNN in optical flow estimation for event cameras [33], which showed significant potential for real-time operations. In addition, SNNs have been applied to depth estimation for event sensors [10], where their method computed the optical flow on the neuromorphic chip and integrated the optical flow with camera velocity to estimate depth.

## 3 Preliminaries

An SNN can be viewed as a "program" for a neuromorphic computer [34]. Formally, an SNN comprises a set of spiking neurons $\mathcal{N} = \{n_i\}_{i=1}^N$ and synapses, where each synapse connects a pair of neurons. The interconnections can be summarized by a matrix $\mathbf{W} = [w_{ij}] \in \mathbb{Z}^{N \times N}$, where the element $w_{ij}$ indicates the strength of connection between $n_i$ and $n_j$, and $w_{ij} = 0$ signifies an absence of connection between the pair. The architecture of an SNN and the model of the spiking neuron define the behavior of the program. Here, we describe the NEBM spiking neuron model (Sec. 3.1) and the NEBM-based SNN for QUBO (Sec. 3.2), adapted from Intel's original implementation [32].

### 3.1 NEBM

NEBM neurons produce outputs probabilistically based on the Boltzmann distribution [32]. An NEBM neuron $n_i$ contains a binary state $s_i \in \{0, 1\}$ that indicates whether the neuron is firing ($s_i = 1$) or not ($s_i = 0$), and an internal state $u_i \in \mathbb{Z}$ that accumulates outputs from connected neurons.

While in theory spiking neurons operate asynchronously, practical neuromorphic computers such as Intel Loihi are fully digital devices [12]. The continuous time dynamics of a spiking neuron are approximated using fixed-size discrete timesteps $t$. It should be reminded that $t$ relates to the algorithmic time of the computation rather than the time of a global synchronous clock.

Based on the algorithmic time formalism, at timestep $t$, an NEBM neuron $n_i$ accumulates inputs from connected neurons into its internal state

$$u_i^{(t)} = u_i^{(t-1)} + \sum_{j \neq i} w_{ij} \Delta s_j^{(t-1)}, \tag{10}$$

where $\Delta s_j^{(t-1)}$ is the output spike of neuron $n_j$ from the previous timestep. The internal state of $n_i$ is then converted to a switching probability

$$P\left(s_i^{(t)} = 1\right) = \frac{1}{1 + e^{u_i^{(t)}/T}}, \tag{11}$$

where $T$ is the *temperature*. If the switching probability (11) exceeds a randomly generated threshold $\theta_i \in [0, 1]$, $s_i$ is set to 1. From here, an output or delta spike is calculated

$$\Delta s_i^{(t)} = s_i^{(t)} - s_i^{(t-1)}. \tag{12}$$

If $\Delta s_i^{(t)}$ takes a non-zero value, *i.e.*, the state $s_i$ changes from the previous timestep, $n_i$ propagates a delta spike to all its connected neurons and enters a *refractory period* for $r_i$ timesteps. Within the refractory period, the binary state $s_i$ remains unchanged. Note that temperature $T$ and length of the refractory period $r_i$ are hyperparameters of NEBM neurons. For more details of NEBM, its hyperparameters and how it is implemented on Loihi 2 given hardware constraints (i.e. no division operator), we refer readers to [32].

### 3.2   NEBM-based SNN for QUBO

The energy encoded by the neuronal states in an SNN is

$$\mathcal{E} = \sum_{i=1}^{N} s_i u_i. \tag{13}$$

Following [32], the NEBM-based SNN algorithm to minimize the energy is summarized in Alg. 2 (see Appendix). Intuitively, the method repeatedly samples and explores the neuronal states guided by the structure of the synapses and evolving state values [21]. The algorithm is executed on the neuromorphic hardware for $M$ algorithmic timesteps, and the state configurations probed at the $M$ timesteps are returned. The state configurations are evaluated off-chip, with the best one returned as the solution. Note that we employ an early version of NEBM where $T$ is kept constant in Alg. 2, though $T$ can be annealed to fine-tune the search. We refer interested readers to [32] for recent advancements of NEBM and the general-purpose NEBM-based QUBO solver.

To enable Alg. 2 to solve QUBO, following [32], we perform the following mapping:

- Assign each binary variable $z_i$ to a neuron $n_i$, and equating the state $s_i$ with the value of $z_i$.
- Associate the quadratic coefficients $\mathbf{Q} = [q_{ij}] \in \mathbb{Z}^{N \times N}$ with the synapse weights $\mathbf{W}$, *i.e.*, $\mathbf{Q} = \mathbf{W}$. This also implies the existence of self connections for the neurons, where $q_{ii} = w_{ii}$ is the strength of the self connection for neuron $n_i$.

Fig. 1b illustrates the mapping. Note that the "mapping" does not imply that $\mathcal{E}$ equates to the QUBO cost; rather, the minimization of $\mathcal{E}$ by Alg. 2 tends to lead to the minimization of the QUBO cost. In this sense, the SNN is a *heuristic* method to solve QUBO.

## 4   Slack-free SNN formulation for HMVC

Instead of converting HMVC to QUBO following the derivations in Problem 2 and applying the SNN described in Sec. 3.2, we develop a novel *slack-free* SNN that directly solves HMVC. Referring to the 0-1 ILP (6), we construct our SNN, named *SF-HMVC*, as follows:

- As before, each binary variable $z_i$ is encoded as the state $s_i$ of an NEBM neuron $n_i$.
- Each $k$-th constraint $\mathbf{b}^{(k)^T} \mathbf{z} \geq 1$ is represented by a neuron $m_k$, whose dynamics are governed by a custom neuron model called *feedback* or *FB* (more details below).

**Algorithm 1** Our proposed SF-HMVC (**note:** the algorithm is executed for each NEBM neuron $n_i$ and FB neuron $m_k$ on the neuromorphic hardware in a parallel way; see [13] for details).

---

**Require:** Co-occurrence matrix $\mathbf{F}$, adjacency matrix $\mathbf{A}$, temp. $T$ and refract. period $r_i$.

1: Initialize $s_i^{(0)} \leftarrow 0$, $\Delta s_i^{(0)} \leftarrow 0$, $u_i^{(0)} \leftarrow -f_{ii}$, refract_counter$_i^{(0)} \leftarrow 0$

2: **for** each timestep $t$ **do**

3: $\quad u_i^{(t)} \leftarrow u_i^{(t-1)} + \sum\limits_{j \neq i}^{N} f_{ij} \Delta s_j^{(t-1)} + \sum\limits_{k=1}^{K} a_{ik} \Delta c_k^{(t-1)}$

4: $\quad p_i^{(t)} \leftarrow \dfrac{1}{1 + \exp(u_i^{(t)}/T)}$

5: $\quad \theta_i \leftarrow rand(0, 1)$

6: $\quad$ **if** neuron $n_i$ is not in refractory period **then**

7: $\qquad$ **if** $p_i^{(t)} \geq \theta_i$ **then**

8: $\qquad\quad s_i^{(t)} \leftarrow 1$

9: $\qquad$ **else**

10: $\qquad\quad s_i^{(t)} \leftarrow 0$

11: $\quad$ **else**

12: $\qquad s_i^{(t)} \leftarrow s_i^{(t-1)}$

13: $\qquad$ refract_counter$_i^{(t)} \leftarrow \max(\text{refract\_counter}_i^{(t-1)} - 1, 0)$

14: $\quad \Delta s_i^{(t)} \leftarrow s_i^{(t)} - s_i^{(t-1)}$

15: $\quad$ send $\Delta s_i^{(t)}$ to connected neurons

16: $\quad$ **if** $\Delta s_i^{(t)} \neq 0$ **then**

17: $\qquad$ neuron $n_i$ enters refractory period

18: $\qquad$ refract_counter$_i^{(t)} \leftarrow r_i$

$\left. \right\}$ NEBM neuron

---

**Require:** Adjacency matrix $\mathbf{A}$.

1: **for** each timestep $t$ **do**

2: $\quad v_k^{(t)} \leftarrow \sum\limits_{i=1}^{N} a_{ik} s_i^{(t)}$

3: $\quad$ **if** $v_k^{(t)} = 0$ **then**

4: $\qquad c_k^{(t)} \leftarrow 1, \Delta c_k^{(t)} \leftarrow -1$

5: $\quad$ **else**

6: $\qquad$ **if** $c_k^{(t-1)} = 0$ **then**

7: $\qquad\quad c_k^{(t)} \leftarrow 0, \Delta c_k^{(t)} \leftarrow 0$

8: $\qquad$ **else**

9: $\qquad\quad c_k^{(t)} \leftarrow 0, \Delta c_k^{(t)} \leftarrow 1$

10: $\quad$ send $\Delta c_k^{(t)}$ to connected neurons

$\left. \right\}$ FB neuron

---

- The weight matrix of our SNN which consists of $N + K$ neurons $[s_1, \ldots, s_N, m_1, \ldots, m_K]$ is

$$\mathbf{W} = \begin{bmatrix} \mathbf{F} & \mathbf{A} \\ \mathbf{A}^T & \mathbf{0}_{K \times K} \end{bmatrix} \in \mathbb{Z}^{(N+K) \times (N+K)}, \tag{14}$$

where $\mathbf{0}_{K \times K}$ is a $K \times K$ matrix of zeros,

  - $\mathbf{A} = [\mathbf{b}^{(1)} \cdots \mathbf{b}^{(K)}] \in \{0, 1\}^{N \times K}$ is the adjacency matrix between NEBM and FB neurons (intuitively, the $k$-th FB neuron $m_k$ is connected to the NEBM neurons corresponding to $z_i$'s that appear in the $k$-th constraint); and

  - $\mathbf{F} = [f_{ij}] = \mathbf{A}\mathbf{A}^T \in \mathbb{Z}^{N \times N}$ is the co-occurrence matrix of the NEBM neurons (intuitively, $f_{ij}$ is high if $z_i$ and $z_j$ appear frequently together in the constraints, while $f_{ii}$ is high if $z_i$ appear in many constraints).

Fig. 1c illustrates the proposed SNN construction, while Alg. 1 summarizes the associated algorithm.

Note that Alg. 1 involves two types of neurons that implement different dynamics, and the neurons are executed in parallel on the neuromorphic hardware [13]. The internal state for $m_k$ is

$$v_k^{(t)} = \sum_{i=1}^{N} a_{ik} s_i^{(t)} \tag{15}$$

where $a_{ik}$ is the element of $i$-th row and $k$-th column of matrix $\mathbf{A}$. An FB neuron $m_k$ is deterministically activated (*i.e.*, its binary state $c_k$ becomes 1) whenever the connected NEBM neurons are *all inactive* (*i.e.*, $v_k = 0$), and vice versa. Then, a negative spike $\Delta c_k$ is generated if the value of $c_k$ is 1, so as to send excitatory signals to the connected NEMB neurons to attempt to satisfy the constraint; otherwise no spikes are generated. The point here is not to over-excite NEBM neurons. Then, the internal state of an NEBM neuron $n_i$ accumulates inhibitory signals from the connected NEBM neurons and excitatory signals from the connected FB neurons, *i.e.*,

$$u_i^{(t)} = u_i^{(t-1)} + \sum_{j \neq i}^{N} f_{ij} \Delta s_j^{(t-1)} + \sum_{k=1}^{K} a_{ik} \Delta c_k^{(t-1)}. \tag{16}$$

Similar to Alg. 2, Alg. 1 is executed on the neuromorphic hardware for $M$ algorithmic timesteps, and the best state configuration probed and evaluated off-chip is taken as the solution. Note that the probing mechanism incurs significant overhead in the execution time of current implementation [10], and an on-chip evaluation strategy could be implemented to address this bottleneck [24, 32].

**Scalability analysis**   Based on the construction of the QUBO-based SNN and the proposed SF-HMVC, we can model the number of spiking neurons required to encode each SNN:

- $n_{\text{neurons}}(\text{QUBO}) = N + (r - 1)K$
- $n_{\text{neurons}}(\text{SF-HMVC}) = N + K$

As $N, K$ and $r$ collectively define the size of the input problem, we can conclude that SF-HMVC scales linearly whereas the QUBO-based SNN scales quadratically with respect to the problem size.

## 5   Experiments

**Implementation**   Our method SF-HMVC was implemented on the Loihi assembly language through Lava framework [2] and deployed on the neuromorphic chip of Intel Loihi 2 [30] with $M = 1000$ algorithmic timesteps (denoted as `SF-HMVC-Loihi`). Using grid search, we selected temperature $T$ from the range $[2, 4, 8, 16, 24, 32]$, refractory period $r_i$ from the range $[8, 16, 32, 64, 128]$ (note that all NEBM neurons were given the same refractory period). We set $\lambda = 2$ for QUBO ( 4) (same as [11]) and grid search $\lambda = [1, 10, 100, 1000]$ for QUBO ( 9). Note that we were unable to change the seed of the random number generator on Intel Loihi 2, which prevented us from repeatedly executing SNNs to obtain error bars in the solution quality. Also, we noticed that energy consumption and runtime of our method and all competitors remained fairly consistent across different runs.

### 5.1   MVC

**Dataset**   We employed the DIMACS benchmark dataset [20]. DIMACS includes graph instances that are both synthetically generated and sourced from real-world applications, including coding theory, fault diagnosis, Keller's conjecture, *etc*. The original DIMACS graphs were for maximum clique problems; we converted these to MVC instances by taking the complement graphs. Only 15 MVC instances where the corresponding SNN for SF-HMVC could fit on Loihi 2 were selected.

**Competitors**   `SF-HMVC-Loihi` was compared to the following methods:

- `ILP-CPU`: MVC was solved using the ILP (3) with Gurobi [18] and evaluated on CPU.
- `QUBO-CPU`: MVC was solved using QUBO (4) with Gurobi [18] and evaluated on CPU. We set $\lambda = 2$ (same as [11]).
- `QUBO-Loihi`: MVC was solved using QUBO (4) with NEBM-based SNN (see Sec. 3.2) and evaluated on Intel Loihi 2. We set $T = 16, r_i = 64, \lambda = 2$.

Table 1: Solution quality $\|\mathbf{z}\|_1$ of the MVC methods for all DIMACS MVC instances.

| Instance | $|V|$ | $|E|$ | ILP-CPU | QUBO-CPU | QUBO-Loihi | SF-HMVC-Loihi |
|---|---|---|---|---|---|---|
| C125.9 | 125 | 787 | 91 | 91 | 122 | 116 |
| C250.9 | 250 | 3141 | 206 | 206 | 239 | 240 |
| gen200_p0.9_44 | 200 | 1990 | 156 | 156 | 185 | 199 |
| gen200_p0.9_55 | 200 | 1990 | 145 | 145 | 187 | 191 |
| hamming6-2 | 64 | 192 | 32 | 32 | 51 | 32 |
| hamming6-4 | 64 | 1312 | 60 | 60 | 64 | 61 |
| hamming8-2 | 256 | 1024 | 128 | 128 | 236 | 128 |
| johnson8-2-4 | 28 | 168 | 24 | 24 | 28 | 24 |
| johnson8-4-4 | 70 | 560 | 56 | 56 | 59 | 69 |
| johnson16-2-4 | 120 | 1680 | 112 | 112 | 120 | 119 |
| keller4 | 171 | 5100 | 160 | 160 | 171 | 170 |
| MANN_a9 | 45 | 72 | 29 | 29 | 38 | 37 |
| MANN_a27 | 378 | 702 | 252 | 252 | 352 | 324 |
| san200_0.9_1 | 200 | 1990 | 130 | 130 | 178 | 199 |
| sanr200_0.9 | 200 | 2037 | 158 | 158 | 184 | 199 |

Table 2: Runtime (in seconds) and energy usage (in Joules) of MVC methods. Note that $\infty$ indicates that the energy was too high for pyJoules to measure.

| Instance | ILP-CPU | | QUBO-CPU | | QUBO-Loihi | | SF-HMVC-Loihi | |
|---|---|---|---|---|---|---|---|---|
| | Time | Energy | Time | Energy | Time | Energy | Time | Energy |
| C125.9 | 0.37 | 32.41 | 1.00 | 94.53 | 3.79 | 0.09 | 3.77 | 0.05 |
| C250.9 | >3600 | 144380.56 | >3600 | $\infty$ | 7.36 | 0.26 | 7.37 | 0.18 |
| gen200_p0.9_44 | 0.1 | 8.92 | 4.60 | 387.83 | 6.03 | 0.09 | 5.18 | 0.14 |
| gen200_p0.9_55 | 0.03 | 2.12 | 2.59 | 203.84 | 5.98 | 0.04 | 5.17 | 0.01 |
| hamming6-2 | 0 | 0.07 | 0.01 | 0.46 | 2.09 | 0.02 | 1.84 | 0.06 |
| hamming6-4 | 0.03 | 2.25 | 0.36 | 25.83 | 2.91 | 0.07 | 1.83 | 0.06 |
| hamming8-2 | 0 | 0.19 | 0.03 | 1.53 | 7.64 | 0.04 | 7.55 | 0.04 |
| johnson8-2-4 | 0.01 | 0.41 | 0.05 | 3.16 | 1.06 | 0.08 | 0.94 | 0.02 |
| johnson8-4-4 | 0 | 0.15 | 0.26 | 18.83 | 2.26 | 0.02 | 1.99 | 0.05 |
| johnson16-2-4 | 0.5 | 3.74 | 4.46 | 316.75 | 3.71 | 0.03 | 3.72 | 0.05 |
| keller4 | 1.14 | 114.13 | 1628.92 | 185173.36 | 5.13 | 0.11 | 5.15 | 0.03 |
| MANN_a9 | 0 | 0.97 | 0.02 | 2.24 | 1.55 | 0.07 | 1.35 | 0.01 |
| MANN_a27 | 0.18 | 12.06 | 0.82 | 72.92 | 10.95 | 0.31 | 9.58 | 0.09 |
| san200_0.9_1 | 0.01 | 0.87 | 0.27 | 22.70 | 5.99 | 0.25 | 5.28 | 0.04 |
| sanr200_0.9 | 77.92 | 10111.06 | 163.98 | 20591.27 | 5.95 | 0.18 | 5.17 | 0.3 |

All hyperparameters of the competitors were selected using grid search. The configuration that demonstrated consistent performance across all problem instances was selected for each method. For SF-HMVC-Loihi, we set $T = 4, r_i = 8$.

**Metrics** Solution quality, runtime (in seconds), and energy consumption (in Joules) were reported for all methods. Runtime and energy consumption on Intel Loihi 2's Oheo Gulch board were measured through the built-in profiler of Lava-Loihi v0.6.0 extension. Runtime on CPU was recorded using built-in functions of Gurobi and the energy usage was recorded using pyJoules [4] on a workstation with an Intel Core i7-11700K CPU @ 3.6GHz and 32GB RAM running Ubuntu 20.04.6 LTS.

**Results** Tabs. 1 and 2 display the results. Overall, SF-HMVC-Loihi was comparable to QUBO-Loihi in all three aspects: solution quality, runtime, and energy usage (the discrepancy between them will be clearer in the next experiment). ILP-CPU and QUBO-CPU outperformed SF-HMVC-Loihi in terms of solution quality (see Tab. 1), since both ILP-CPU and QUBO-CPU were globally optimal methods. The equal solution quality of the global methods also indicated the existence of a suitable penalty weight for QUBO (4) in the grid search range of $\lambda$. On runtime (see Tab. 2), though ILP-CPU and QUBO-CPU solved many instances in under 1 s, a few instances took them minutes to hours to solve. In contrast, SF-HMVC-Loihi consistently solved all instances within 10 s. Also, on energy usage, our method significantly outperformed ILP-CPU and QUBO-CPU.

Table 3: Solution quality $\|\mathbf{z}\|_1$ of the HMVC methods for all synthetic HMVC instances. Infeasible solutions are indicated in red, with number of constraint violations in brackets. N/A means that the SNN was not able to be embedded into the Loihi 2 due to capacity limitations.

| Instance | $|V|$ | $|F|$ | ILP-CPU | QUBO-CPU | QUBO-Loihi | SF-HMVC-Loihi |
|---|---|---|---|---|---|---|
| 3-uniform_HMVC01 | 30 | 981 | 3 | 3 | 3 (68) | 3 |
| 3-uniform_HMVC02 | 30 | 2129 | 9 | 9 | 19 (1214) | 9 |
| 3-uniform_HMVC03 | 30 | 2888 | 15 | 15 | N/A | 15 |
| 3-uniform_HMVC04 | 30 | 3327 | 22 | 30 | N/A | 29 |
| 3-uniform_HMVC05 | 50 | 3506 | 5 | 5 | N/A | 5 |
| 3-uniform_HMVC06 | 50 | 6538 | 8 | 8 | N/A | 8 |
| 3-uniform_HMVC07 | 50 | 7333 | 10 | 50 | N/A | 10 |
| 3-uniform_HMVC08 | 70 | 7326 | 4 | 4 | N/A | 4 |
| 3-uniform_HMVC09 | 100 | 5979 | 20 | 20 | N/A | 20 |
| 3-uniform_HMVC10 | 200 | 7430 | 10 | 10 | N/A | 19 |
| 4-uniform_HMVC11 | 30 | 4914 | 3 | 15 | N/A | 5 |

## 5.2 HMVC

**Dataset** We generated 11 synthetic HMVC instances as follows: first, a set of $N$ vertices $V$ were created, then a set of $K$ degree-$r$ hyperedges $F$ were randomly generated. The values of $N$, $K$ and $r$ were selected to ensure that the SNN (for SF-HMVC) could fit on Loihi 2.

**Competitors** SF-HMVC-Loihi was compared to the following methods:

- ILP-CPU: HMVC was solved using the ILP Eq. (6) with Gurobi [18] and evaluated on CPU.
- QUBO-CPU: HMVC was solved using QUBO Eq. (9) with Gurobi [18] and evaluated on CPU. We set $\lambda = 10$.
- QUBO-Loihi: HMVC was solved using QUBO Eq. (9) with NEBM-based SNN (see Sec. 3.2) and evaluated on Intel Loihi 2 [30]. We set $T = 32, r_i = 8, \lambda = 10$.

All hyperparameters of these competitors were selected using grid search. The configuration that demonstrated consistent performance across all problem instances was selected for each method. For SF-HMVC-Loihi, we set $T = 8, r_i = 8$.

**Results** Tabs. 3 and 4 display the results. As expected, ILP-CPU outperformed SF-HMVC-Loihi in solution quality, since ILP-CPU was a global method. Interestingly, our method occasionally found better solutions than QUBO-CPU. This was probably because, as the problem difficulty (number of variables) increases, it became more challenging for QUBO-CPU. Note that SF-HMVC-Loihi handled all instances reasonably well. In contrast, QUBO-Loihi either could not find feasible solutions, or correspoding SNN could not be embedded onto Loihi 2. This suggests that the introduction of slack variables in QUBO-Loihi made the search space and/or problem size too large.

As presented in Table 4, while ILP-CPU performed better than SF-HMVC-Loihi in terms of runtime, our method was more efficient in terms of energy consumption. Furthermore, our method significantly surpassed QUBO-CPU and QUBO-Loihi in both runtime and energy efficiency.

## 6 Limitations and conclusions

Several limitations of our work can be identified:

L1: The capacity of the neuromorphic computer available to us (a single Intel Loihi 2 chip [30] which has 128 neuromorphic cores) was relatively low, which prevented testing of large problem instances.
L2: There is a lack of public benchmarks on HMVC problems. L1 and L2 together precluded an assessment of the generalizability of the methods more practical problem instances.
L3: Changing the seed of the random number generator on Intel Loihi 2 was inaccessible to us, which precluded error bars in solution quality.

Table 4: Runtime (in seconds) and energy usage (in Joules) of HMVC methods. Note that $\infty$ indicates that the energy was too high for pyJoules to measure, while `N/A` indicates the instance could not be embedded into Loihi 2 due to capacity limitations.

| Instance | ILP-CPU | | QUBO-CPU | | QUBO-Loihi | | SF-HMVC-Loihi | |
|---|---|---|---|---|---|---|---|---|
| | **Time** | **Energy** | **Time** | **Energy** | **Time** | **Energy** | **Time** | **Energy** |
| 3-uniform_HMVC01 | 0.00 | 0.24 | 198.38 | 14748.67 | 57.82 | 18.89 | 1.02 | 0.04 |
| 3-uniform_HMVC02 | 0.01 | 0.27 | 1464.06 | 110916.35 | 174.32 | 269.43 | 1.00 | 0.02 |
| 3-uniform_HMVC03 | 0.09 | 4.64 | 21.89 | 1548.77 | N/A | N/A | 1.00 | 0.04 |
| 3-uniform_HMVC04 | 0.26 | 15.79 | >3600 | 27582.99 | N/A | N/A | 1.02 | 0.01 |
| 3-uniform_HMVC05 | 0.01 | 0.58 | >3600 | $\infty$ | N/A | N/A | 1.49 | 0.07 |
| 3-uniform_HMVC06 | 0.02 | 1.03 | >3600 | 33883.58 | N/A | N/A | 1.78 | 0.08 |
| 3-uniform_HMVC07 | 0.03 | 1.52 | >3600 | 54487.73 | N/A | N/A | 1.52 | 0.16 |
| 3-uniform_HMVC08 | 0.02 | 0.67 | >3600 | $\infty$ | N/A | N/A | 2.32 | 0.05 |
| 3-uniform_HMVC09 | 0.02 | 0.66 | >3600 | $\infty$ | N/A | N/A | 3.18 | 0.03 |
| 3-uniform_HMVC10 | 0.02 | 2.50 | >3600 | 19186.52 | N/A | N/A | 6.03 | 1.38 |
| 4-uniform_HMVC11 | 0.02 | 0.45 | >3600 | 106312.87 | N/A | N/A | 1.18 | 0.03 |

L4: SNNs are ultimately heuristic algorithms, which complicate theoretical analyses on solution quality and runtime complexity.

Despite the limitations above, the results showed clear trends of the greater scalability of the proposed method SF-HMVC, in that it was able to solve HMVC problem instances where the existing method could not. Moreover, SF-HMVC on Loihi 2 exhibited measurably lower energy consumption than global solvers on CPU, further supporting neuromorphic computing as an energy-efficient alternative.

## Acknowledgments and Disclosure of Funding

We would like to acknowledge Intel Labs and the Intel Neuromorphic Research Community (INRC) for providing access to Loihi 2. We thank Intel's Neuromorphic Computing Lab (NCL) for developing NEBM algorithm and NEBM-based QUBO solvers, which was central to our work. We thank Philipp Stratmann from NCL for the documentation related to NEBM. We also thank all the members of NCL for their support on technical issues. Tat-Jun Chin is SmartSat CRC Professorial Chair of Sentient Satellites.

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

# Appendix

---

**Algorithm 2** NEBM-based SNN for energy minimization (**note:** the algorithm is executed for each neuron $n_i$ on the neuromorphic hardware in a parallel way; see [13] for details).

---

**Require:** Weight matrix $\mathbf{W}$, temperature $T$ and length of refractory period $r_i$.

1: Initialize $s_i^{(0)} \leftarrow 0$, $\Delta s_i^{(0)} \leftarrow 0$, $u_i^{(0)} \leftarrow -w_{ii}$, refract_counter$_i^{(0)} \leftarrow 0$
2: **for** each timestep $t$ **do**
3:     $u_i^{(t)} \leftarrow u_i^{(t-1)} + \sum\limits_{j \neq i}^{N} w_{ij} \Delta s_j^{(t-1)}$
4:     $p_i^{(t)} \leftarrow \dfrac{1}{1 + \exp(u_i^{(t)}/T)}$
5:     $\theta_i \leftarrow rand(0, 1)$
6:     **if** neuron $n_i$ is not in refractory period **then**
7:         **if** $p_i^{(t)} \geq \theta_i$ **then**
8:             $s_i^{(t)} \leftarrow 1$
9:         **else**
10:            $s_i^{(t)} \leftarrow 0$
11:     **else**
12:         $s_i^{(t)} \leftarrow s_i^{(t-1)}$
13:         refract_counter$_i^{(t)} \leftarrow \max($refract_counter$_i^{(t-1)} - 1, 0)$
14:     $\Delta s_i^{(t)} \leftarrow s_i^{(t)} - s_i^{(t-1)}$
15:     send $\Delta s_i^{(t)}$ to connected neurons
16:     **if** $\Delta s_i^{(t)} \neq 0$ **then**
17:         neuron $n_i$ enters refractory period
18:         refract_counter$_i^{(t)} \leftarrow r_i$

---

