# OpenReview forum: "Slack-Free Spiking Neural Network Formulation for Hypergraph Minimum Vertex Cover"
_NeurIPS.cc/2024/Conference — NeurIPS 2024 poster_

### Official Review · Reviewer_9cvo · 2024-06-23

**Soundness:** 2
**Presentation:** 3
**Contribution:** 3
**Rating:** 5
**Confidence:** 4

**Summary:**

Traditional SNN methods for combinatorial optimization necessitate the use of penalty terms with slack variables to maintain feasibility constraints. The paper introduces a novel Spiking Neural Network (SNN) formulation designed to solve the Hypergraph Minimum Vertex Cover (HMVC) problem without requiring slack variables. The proposed SF-HMVC replaces slack variables with additional spiking neurons that check and correct constraints, facilitating convergence to feasible solutions.

**Strengths:**

1. Innovative approach to solving HMVC without slack variables, reducing the search space and improving solver effectiveness.
2. Consistently high-quality solutions across multiple problem instances.
3. The paper structure is well-organized.

**Weaknesses:**

1. The scale of experiments are relatively small and simple.
2. Lack of comparison with other solving methods, e.g. Gurobi and D-Wave.
3. There are some typos. E.g. in line 102, *whot* -> *who*.

**Questions:**

1. How scalable is the proposed SNN method with future advancements in neuromorphic hardware?
2. Can the approach be generalized to other combinatorial optimization problems beyond HMVC?

**Limitations:**

1. The neuromorphic hardware capacity is currently limited, restricting the scale of problem instances that can be tested.
2. There is a lack of public HMVC benchmarks, complicating the evaluation of the method's generalizability.

---

> ### Author Rebuttal · Authors · 2024-08-01
>
> Thanks for the feedback.
>
> 1. Despite the relatively small problem instances that can be solved by the neuromorphic hardware available (Loihi 2), we would like to point out that the selected problem instances were already sufficient to convincingly illustrate the benefit for the proposed approach. Particularly for HMVC, the proposed SNN (SF-HMVC) could provide good results on Loihi 2, while the baseline QUBO-based SNN either returned infeasible results or could not be executed on the hardware due to exceeding resource limits. See Tables 3 and 4.
>
> 2. Note that we have compared against Gurobi solver; see **Competitors** in Sec. 5.1, and columns `ILP-CPU` and `QUBO-CPU` in Tables 1, 2, 3, and 4. Being an established optimization software that guarantees global optimality, the Gurobi-based solutions provide a reference for the quality. Other the other hand, the energy consumption figures in Tables 2 and 4 show that the SNN algorithms, particularly `SF-HMVC` consume at least an order of magnitude less energy than Gurobi.
>
>    Thanks for the suggestion to compare against D-Wave. While both neuromorphic computers and quantum annealers are Ising solvers and thus can solve QUBO, the fact is that solving HMVC on D-Wave will still require adding slack variables to convert HMVC to QUBO. The additional variables will consume the qubit budget, thus reducing the size of the HMVC instances that can be solved using D-Wave. In contrast, the neuromorphic paradigm provides more fiexibility to handcraft SNNs, which we exploited to develop SF-HMVC, which is a slack-free formulation. Nonetheless, we agree that comparing against D-Wave is interesting, which we will leave as future work.
>
> 3. Thank you for pointing out the typos! If accepted, we will carefully check the paper and remove typos.
>
> **How scalable is the proposed SNN method with future advancements in neuromorphic hardware?**
>
>    We have added an analysis of the scalability of SF-HMVC versus the QUBO-based SNN:
>
>    Let $n_{neurons}$ be the number of spiking neurons needed by the SNN, and $N$, $K$ and $r$ respectively be the number of vertices, number of hyperedges, and hyperedge degree/size of the input hypergraph ($r > 2$ for HMVC; see Problem 2). We have:
>
>    $n_{neurons} \text{(QUBO)} = N + (r-1)K$
>
>    $n_{neurons} \text{(SF-HMVC)} = N + K$
>
>    Fig. 1 illustrates the SNN construction. Fundamentally, treating $N$, $K$ and $r$ as input (size) parameters, *SF-HMVC scales linearly* whereas the *QUBO-based SNN scales quadratically* with the HMVC problem size.
>
> **Can the approach be generalized to other combinatorial optimization problems beyond HMVC?**
>
> As surveyed in Sec. 2, the flexibility of the neuromorphic approach allows SNNs to be handcrafted for combinatorial problems. So far, this has included constraint satisfaction problems (in the context of Sudoku [8,29], graph coloring [17], graph hamiltonian [21]) and Boolean satisfiability [21,32].
>
> Our work is the first to conduct combinatorial *optimization* using a handcrafted SNN (previous such works were QUBO-based that employed the generic SNN QUBO solver).
>
> Based on the broader research and our work, there is great potential to generalize the handcrafted approach to other combinatorial optimization problems. Moreover, HMVC is a general problem with many related formulations (set cover, hitting set, transversal) and special cases (MVC, max independent set), hence, our method is applicable to a wide variety of problems.
>
> **The neuromorphic hardware capacity is currently limited**
>
>    Note that while our experiments were conducted on a single-chip system, stackable multiple-chip systems have been developed [A]. We also highlight the recent unveiling of the world's largest neuromorphic system by Intel, which contains "1.15 billion neurons and 128 billion synapses" [1]. With significant commitment by the chipmaking industry on neuromorphic computing, there is great potential for the hardware limitation to be resolved in the near term, thus, it is important for the AI/ML community to focus on SNN algorithm development now.
>
> **Lack of public HMVC benchmarks**
>
> We would like to reiterate point 1 above, where the current datasets/results are already sufficient to convincingly illustrate the much better scalability and performance of the proposed SF-HMVC over the QUBO-based SNN.
>
> [A] Mehonic et al. Roadmap to Neuromorphic Computing with Emerging Technologies. arXiv:2407.02353

---

> > ### Comment · Reviewer_9cvo · 2024-08-09
> >
> > Thanks for authors' rebuttal!
> >
> > Overall I believe this is a technically solid work and I'm keeping my original rating with increased confidence.

---

### Official Review · Reviewer_cGCZ · 2024-07-09

**Soundness:** 2
**Presentation:** 3
**Contribution:** 3
**Rating:** 5
**Confidence:** 3

**Summary:**

This paper presents a novel approach to solving the Hypergraph Minimum Vertex Cover (HMVC) problem using spiking neural networks (SNNs). The authors introduce a slack-free formulation (SF-HMVC) that directly translates the constraints of the HMVC problem into the dynamics of SNN neurons, specifically targeting implementation on neuromorphic hardware such as Intel's Loihi 2. The paper demonstrates that the proposed method can effectively solve HMVC problems and provides a comparative analysis with other optimization techniques, such as Integer Linear Programming (ILP) and Quadratic Unconstrained Binary Optimization (QUBO).

**Strengths:**

1. This paper is well-written.

2. The proposed SF-HMVC approach is designed to be scalable, handling larger problem instances effectively. The parallel nature of neuromorphic computing allows for efficient processing of complex optimization problems.

3. The results show that SF-HMVC can occasionally outperform QUBO-CPU methods in solution quality for larger problem instances, indicating that the SNN-based approach can be competitive with traditional optimization algorithms.

**Weaknesses:**

1. Algorithm 1 is not the algorithm proposed in the article, yet it occupies a significant portion of the paper. It is recommended to move it to the appendix.

2. Could the authors provide a more detailed explanation of the setup and motivation for the W matrix, such as how the A matrix and F matrix are configured?

3. The experiments are influenced by many hyperparameters, such as $\lambda$ and timestep $T$. It is recommended to conduct appropriate ablation experiments.

**Questions:**

Please see weakness.

**Limitations:**

The authors raise some limitaions, for example, the capacity of the Loihi 2 is low, which is limited by the development of hardware.

---

> ### Author Rebuttal · Authors · 2024-08-04
>
> Thanks for the feedback.
>
> 1. We included Algorithm 1 in the paper to make it self-contained, however, if accepted, we will move it to the appendix in the camera-ready version.
>
> 2. The motivation of the $\mathbf{W}$ matrix is to capture the connection strengths between neurons within the SF-HMVC SNN, which can be interpreted as interactions between NEBM-NEBM and NEBM-FB neurons.
>
> - The binary variables in Eq. 6 are encoded as NEBM neurons, which are designed to inhibit each other to facilitate the minimization of the objective function. The connection strengths between NEBM-NEBM neurons are associated with $\mathbf{F}$ matrix, where the entries $f_{ij}$ are calculated based on the occurrence and co-occurrence of variables $z_i$, $z_j$ within the problem constraints.
>
>  - The $\mathbf{A}$ matrix defines the connection between NEBM versus FB neurons. The motivation of the $A$ matrix is to introduce an FB neuron to each set of NEBM neurons that belong to the same constraints. The FB neuron is active only when all NEBM neurons under its "observation" are turned off (corresponding to the constraint being violated) and remains silent otherwise. Once activated, the FB neuron sends excitatory signals until the constraint is satisfied.
>
>    We hope the above clarifies the setup and motivation for $\mathbf{W}$ matrix.
>
> 3. Note that the baseline QUBO-based SNN (Algorithm 1) requires 3 hyperparameters:
>    - $\lambda$ for the weight of the penalty term.
>    - $T$ for the temperature.
>    - $r_i$ for the refractory period.
>
>    The proposed SNN SF-HMVC (Algorithm 2) requires only 2 hyperparameters ($T$ and $r_i$) due to our slack-free formulation. Moreover, since the total number of steps $M$ given to both algorithms is fixed to $1000$, for brevity we do not perform ablation on $M$.
>
>    Figure R3 in the PDF under **Author Rebuttal** shows the ablation studies of the influence of $T$ and $r_i$ on the solution quality of SF-HMVC on Loihi 2. The results show that there are hyperparameter configurations where our method consistently yielded high-quality solutions.
>
>    Also, as mentioned in L194, we have conducted grid search to find common settings of hyperparameters ($\lambda$ for the QUBO formulations, temperature and refractory period for the SNN models) that work well for all problem instances. If accepted, we will define the chosen hyperparameter values in the paper.

---

> > ### Comment · Reviewer_cGCZ · 2024-08-13
> > **Response to the authors**
> >
> > Dear authors,
> >
> > Thanks for your rebuttal. I think the authors addressed my questions and I appreciate the supplemented experiments. So I increase my confidence score.

---

### Official Review · Reviewer_K44F · 2024-07-12

**Soundness:** 2
**Presentation:** 3
**Contribution:** 2
**Rating:** 5
**Confidence:** 1

**Summary:**

The paper presents a novel approach to solving the Hypergraph Minimum Vertex Cover (HMVC) problem using Spiking Neural Networks (SNNs) on neuromorphic hardware, which is a significant contribution to the field of combinatorial optimization in neuromorphic computing. Here's a detailed review based on various aspects of the paper:

**Strengths:**

* The integration of spiking neural networks (SNNs) with quantum-inspired optimization techniques represents a novel approach to solving hard minimum vertex cover (HMVC) problems. This hybrid method leverages the strengths of both neuromorphic computing and quantum mechanics principles, potentially opening new avenues for complex problem-solving.
* The authors provide a clear explanation of the neuromorphic computing background, the limitations of existing SNN approaches, and the rationale behind their novel method. The use of NEBM spiking neurons and the detailed description of the network architecture and dynamics add depth to the technical discussion.

**Weaknesses:**

* The comparison is mainly limited to traditional SNN-based QUBO solvers. Including comparisons with other contemporary optimization techniques, especially those that are non-neuromorphic, could provide a clearer benchmarking against the state-of-the-art in broader combinatorial optimization research.
* While the paper claims improved energy efficiency, detailed metrics or comparative energy consumption data are lacking. Providing explicit energy consumption figures or a more detailed analysis could help substantiate these claims and compare them with other methods' energy profiles.
* Although the paper discusses the potential scalability of the approach, there is limited empirical evidence supporting this claim, especially in terms of larger and more complex problem instances. Detailed scalability analysis, possibly through theoretical modeling or additional experiments, would be beneficial.
* The effectiveness of the proposed method is closely tied to the availability and performance of specific neuromorphic hardware. This dependency could limit its accessibility and practicality, especially in environments where such hardware is not readily available or is cost-prohibitive.

**Questions:**

* Are there any more dataset available for benchmarking?
* What measures have been taken to prevent overfitting in the model during the experimental phase?
* Are there any scalability concerns when applying this method to larger or more complex problem sets?

**Limitations:**

limitations are discussed in the paper

---

> ### Author Rebuttal · Authors · 2024-08-01
>
> Thanks for the feedback.
>
> - Note that we have compared against two contemporary (non-neuromorphic) optimization techniques: integer linear programming (ILP) and quadratic unconstrained binary optimization (QUBO), both implemented using a leading optimization sofware (Gurobi) and executed on an Intel Core i7 CPU. See columns `ILP-CPU` and `QUBO-CPU` in Tables 1 to 4.
>
> - Note that we have included explicit energy consumption figures in Tables 2 and 4 for all methods. The energy measurements were obtained using the built-in profiler in the Loihi framework for the SNN solutions and pyJoules [5] for the CPU solutions.
>
>    To summarize, the SNNs on neuromorphic hardware (`SF-HMVC-Loihi` and `QUBO-Loihi`) consumed significantly less energy (at least 1 order of magnitude less, and often several orders of magnitudes less) than the CPU-based methods (`ILP-CPU` and `QUBO-CPU`).
>
>    Also, from Table 3, our method `SF-HMVC-Loihi` could return good solutions for all HMVC instances, whereas `QUBO-Loihi` was infeasible in a majority of the instances.
>
> - The reviewer makes a good point, thanks! Since the capacity of the current neurmorphic hardware precludes testing on larger and more complex HMVC instances, detailed scalability analysis is useful.
>
>    We conduct this analysis by modeling the number of spiking neurons $n_{neurons}$ needed by either the QUBO-based SNN or the proposed SF-HMVC. Let $N$, $K$ and $r$ respectively be the number of vertices, number of hyperedges, and hyperedge degree/size of the input hypergraph ($r > 2$ for HMVC; see Problem 2 in the paper). We have:
>
>    $n_{neurons} \text{(QUBO)} = N + (r-1)K$
>
>    $n_{neurons} \text{(SF-HMVC)} = N + K$
>
>    Fig. 1 illustrates the SNN construction. Since $N$, $K$ and $r$ collectively define the size of the input, we can conclude that *SF-HMVC scales linearly* whereas the *QUBO-based SNN scales quadratically* with the problem size. Observe the HMVC results in Tables 3 and 4 where QUBO-SNN failed to be embedded in Loihi 2 due to exceeding the resource limits, whereas SF-HMVC could still find high-quality solutions. If accepted, we will add the above analysis.
>
> - Note that the SNN formalism [31] is independent of neuromorphic hardware implementation, hence, SNN algorithm design and comparative analysis can be abstracted from the hardware, e.g., see above on $n_{neurons}$.
>
>    While empirical validation is tied to the hardware, we note that practical neuromorphic hardware has become very accessible in recent years. For example, academic researchers can sign up at no cost to the Intel Neuromorphic Research Community to obtain free cloud-based access to Loihi 2. Smaller firms are also offerring mature neuromophic hardware at affordable prices, e.g., see BrainChip's Akida™ PCIe Board.
>
> **Are there any more dataset available for benchmarking?**
>
> There are other datasets for benchmarking (e.g., BHOSLIB for MVC). However, the selected problem instances were already sufficient to convincingly illustrate the benefit for the proposed approach. Particularly for HMVC, the proposed SNN (SF-HMVC) could provide good results on Loihi 2, while the baseline QUBO-based SNN either returned infeasible results or could not be executed on the hardware due to exceeding resource limits. See Tables 3 and 4.
>
> While testing on larger instances will require larger-scale hardware, note that Intel recently unveiled the world's largest neuromorphic system, containing "1.15 billion neurons and 128 billion synapses" [1]. Thus, it is important for the AI/ML community to focus on SNN algorithm development now.
>
> **What measures have been taken to prevent overfitting in the model during the experimental phase?**
>
> Note that our SNN directly optimizes solutions for the combinatorial problem without involving separate training and inference phases. An interpretation of preventing "overfitting" in our case can be finding hyperparameter settings that generally work for all instances.
>
> As mentioned in L213, for each method, we conducted grid search for hyperparameters and selected a single configuration that demonstrated consistent performance across all problem instances. If accepted, we will further elaborate this point.
>
> **Are there any scalability concerns when applying this method to larger or more complex problem sets?**
>
> In the analysis above, we have shown that our SNN (SF-HMVC) scales linearly while the baseline QUBO-based SNN scales quadratically, hence, our method is provably more scalable. The fact that SF-HMVC does not need to optimize slack variables will also allow it to return higher quality results due to simpler loss landscapes.
>
> The current main limitation is due to the hardware, which, as highlighted above, is on the way to be resolved judging by recent major developments (see above).

---

### Official Review · Reviewer_Que6 · 2024-07-14

**Soundness:** 3
**Presentation:** 2
**Contribution:** 3
**Rating:** 8
**Confidence:** 4

**Summary:**

The paper presents a method that solves a specific type of problem of combinatorial optimization (hypergraph minimum vertex cover) through spiking neural networks. The method, tested on small versions of the problem, enables a neuromorphic hardware system made by Intel to arrive at a result in cases where previous methods did not, and with less energy consumption than certain other methods if they are ran on CPU.

**Strengths:**

The work advances the field of neural networks beyond machine learning, i.e. in combinatorial optimization.

Moreover, it concerns spiking neural networks, a field that has been attracting growing interest.

Furthermore, the method is actually tested in neuromorphic hardware, contrary to many papers in the field that only include theoretical implications for neuromorphic hardware.

Moreover, the work presents new partial evidence that neuromorphic algorithms and neuromorphic hardware may have advantages over more conventional approaches, a promise of this research field that has been long looking for fulfillment.

**Weaknesses:**

On the other hand, the paper has several weaknesses.

1. The paper does not make clear how significant the type of problem addressed here (HMVC) is, and why it is significant.

2. A figure illustrating an example toy problem of HMVC as well as its solution in the QUBO-based SNN and in the newly proposed method would be very helfpul in clarifying the paper's contribution.

3. The literature review around neuromorphic hardware is rather narrow, and largely focuses on Loihi alone. For example, even narrowly focusing on hardware for Ising models, here is a review of various implementations that could be cited: https://www.nature.com/articles/s42254-022-00440-8

4. Section 2 cites works where SNNs have performed well, but only does this for tasks outside of machine learning. For machine learning there is only a pointer to a survey, which is related to Loihi again, and again focuses largely beyond machine learning. This should be mitigated, especially because in reality, machine learning is arguably the more popular application of SNNs, and spiking machine learning models have in fact outperformed non-spiking ones concretely and under fair hardware conditions in certain cases. Here are the two examples that I am aware of: https://arxiv.org/abs/2009.06808 (under certain temporal dynamics SNNs were shown to be theoretically optimal and practically surpassed ANNs in accuracy) and https://openreview.net/forum?id=iMH1e5k7n3LI (spikes improved inference speed without accuracy drops, and even on GPUs).

5. Most importantly, I believe that the paper's contribution to the broader field of Neural Networks might not be significant, for the following reasons.

- 5a. The work is specifically related to SNNs alone, and specifically related to their use for combinatorial optimization and even more narrowly, specifically HMVC. That is a rather niche scenario.
- 5b. The results are on rather small scale demonstrations.
- 5c. It is unclear that there is any advantage from the neuromorphic aspect. Specifically, the presented heuristic-based algorithm (or a suitable adaptation) has not been tested on CPU. It seems that in the same way that previous SNN approaches were comparable to QUBO and could thus be run on CPU, there must be an analog of the new method that can also be tested on CPU, and might be more energy efficient than Loihi. After all, QUBO on CPU is more efficient than QUBO on Loihi, as the paper shows. Similarly, is QUBO on CPU the baseline to beat to claim a neuromorphic advantage, or should it be eg a microcontroller or an FPGA?

6. The authors mention that they could not change the random seed to obtain statistics. For a stochastic algorithm like the one presented, this seems rather important. Could this be mitigated, eg by running the algorithm on CPU and obtaining some statistics there?

**Questions:**

Could the authors address the above points?

Is there an intuitive explanation why QUBO on Loihi is less efficient than on CPU, and does this explanation not apply to the authors' new approach?

Why can't the random seed be changed?

**Limitations:**

Some of the limitations mentioned in this review are mentioned in the paper, but not all.

---

> ### Author Rebuttal · Authors · 2024-08-01
>
> Thanks for the feedback.
>
> 1. We have indicated in L79 the practical applications of HMVC in "computational biology [9], computer network security [19], resource allocation [7] and social network analysis [23]." More fundamentally, HMVC is a general problem with many related formulations (set cover, hitting set, transversal, MVC, max independent set) [A], hence, HMVC algorithms have wide applicability. In short, HMVC is a significant problem and a good SNN algorithm for it will have major impact. If accepted, we will expand on the above.
>
> 2. Fig. 1 illustrates a toy problem and the previous QUBO-based SNN and our handcrafted SNN (SF-HMVC). If accepted, we will update the figure with solutions from both methods.
>
> 3. Note that our focus is on *SNN algorithms for optimization* (L39). In L29 we have touched upon IBM TrueNorth [25] and Intel Loihi [12,28] which are the two major neuromorphic hardware that have supported the development and experimentation of SNN-based optimization [6, 10, 11, 13, 24, 26, 28, 30, 32].
>
>    We agree that highlighting other potential implementations is useful. If accepted, we will cite the Nature Reviews Physics paper suggested by the reviewer, which covers diverse technologies at various stages of maturity such as spintronics, memristors, quantum annealers, etc.
>
> 4. Since our focus is on SNN for optimization, Sec. 2 mainly surveyed works related to that. Nevertheless, we agree that the section could include major works on SNN for machine learning. If accepted, we will include the references.
>
> 5. Our focus on SNN for optimization is closely aligned with the Primary Area of **Optimization (e.g., convex and non-convex, stochastic, robust)** in NeurIPS 2024.
>
> - 5a. HMVC is a fundamental problem with wide applicability---we have justified this in point 1 above.
>
>    Second, as the reviewer said, SNN "... has been attracting growing interest". Combinatorial optimization is a major strand of research in SNN [8,11,13,17,21,22,26,32]. Moreover, the Nature Reviews Physics paper cited by the reviewer surveys numerous hardware implementations to *solve combinatorial optimization*---clearly this indicates the importance of research into novel hardware and algorithms for combinatorial optimization.
>
> - 5b. Note that we have comprehensively evaluated the algorithm on the DIMACS benchmark [20] for MVC and synthetic instances for HMVC. As stated in L206, only instances that fit on Loihi 2 could be tested. Nonetheless, small HMVC instances (Tables 3 and 4) were already sufficient to illustrate the superiority of our `SF-HMVC-Loihi` over the previous `QUBO-Loihi`.
>
>    While the current capacity of Loihi 2 limits the problem size (which also affected [10, 13, 24, 28, 32]), our careful formulations and rigorous benchmarking provide a clear indication of the potential of our approach. Indeed, the reviewer counted "actual testing in neuromorphic hardware" as a strength.
>
>    See also our responses to K44F and 9cvo on scalability.
>
> - 5c. The reviewer suggested the existence of a CPU analog of our new method that "might be more energy efficient than Loihi", but did not provide details of this algorithm.
>
>    Note that the previous SNN is QUBO-based and hence has a direct CPU analog. In contrast, our method is a *handcrafted* SNN for HMVC that is not QUBO-based. It is unclear what the CPU analog of our method is.
>
>    Second, the reviewer claimed that our paper showed that "QUBO on CPU is more efficient than QUBO on Loihi" (based on the context, we presume this meant *energy efficiency*). Note that all our results (Tables 2 and 4) point to a much higher energy consumption by the CPU solutions (`QUBO-CPU` and `ILP-CPU`) than the SNN solutions (`QUBO-Loihi` and `SF-HMVC-Loihi`) ==> The reviewer probably misread the results.
>
>    We believe CPU is the correct baseline since it is the currently dominant hardware for combinatorial optimization. Comparisons with microcontrollers and FPGAs are also interesting, which we will conduct as future research; thanks for the suggestion!
>
> 6. To obtain statistics of results, we simulated the SNNs on CPU via the Lava Software Framework; see `QUBO-Lava` and `SF-HMVC-Lava` in the PDF under **Author Rebuttal**. Note that Lava simulates asynchronous processing on CPU and hence the performance (particulary the energy consumption) does not closely reflect the performance on neuromorphic hardware.
>
>    It is also important to note that the Lava versions are not the *intrinsic* CPU analogs of the SNN methods.
>
> **Why QUBO on Loihi is less efficient than on CPU?**
>
> Again, we believe the reviewer has mistaken, since our results (Tables 2 and 4) point to a much higher energy consumption by `QUBO-CPU` than `QUBO-Loihi`.
>
> Our method `SF-HMVC-Loihi` also consumed much less energy than `QUBO-CPU` and `ILP-CPU`.
>
> **Why can't the random seed be changed?**
>
> The current Intel API that was available to us did not include functionality to change the random seed on Loihi 2. However, note the additional results on Lava simulation mentioned above.
>
> **Summary**
>
> The reviewer did not report technical flaws. The main concerns seem to be on the relevance and significance of the contribution, which we have adequately addressed. We hope our clarifications on the experiments and results further demonstrate the significance of our findings.
>
> [A] Wikipedia: Vertex cover in hypergraphs

---

> ### Comment · Reviewer_Que6 · 2024-08-09
>
> I would like to thank the authors for their response.
> It is helpful. However, some important parts remain partly unclear.
>
> My understanding that QUBO on Loihi is less efficient than on CPU was based on the fact that except for its smallest versions, the problem did not fit on Loihi.
> The Loihi 2 board that was used, as far as I understand, has 128 cores, whereas the compared Intel Core i7-11700K CPU has only 8. Also, both chips have about the same number of transistors. This seems to be a type of inefficiency on the Loihi side.
>
> Based on this context, I will rephrase and expand on some of my previous questions that I found to be only partly addressed by the authors.
>
> - What causes this hardware inefficiency? In other words, what is Loihi's hardware bottleneck for larger problems, be it with QUBO or with SF-HMVC, and is it a fundamental weakness of neuromorphic computing in general, or rather of the specific implementation that is Loihi?
> - Is there a fundamental reason why SF-HMVC cannot have an analogue that would work on CPU? Essentially, is there a neuromorphic principle that is exploited for this approach but cannot be used with CPUs?
> - I suspect that a hypothetical SF-HMVC-CPU implementation would not only be more efficient than QUBO (CPU or Loihi), but would also allow for larger problems with fewer hardware resources than SF-HMVC-Loihi. This suspicion is based on the comparison between QUBO-CPU and QUBO-Loihi. Is this suspicion wrong, and, if so, why?

---

> ### Author Response · Authors · 2024-08-10
>
> As outlined in the abstract, our **claimed contribution** is a scalable handcrafted SNN called SF-HMVC for solving HMVC on neuromorphic computers. The claim was justified via scalability analysis and experiments on Loihi 2 that compared SF-HMVC against the previous QUBO-based SNN. The results on Loihi 2 also showed that SF-HMVC was more *energy-efficient* than the CPU solutions.
>
> It seems Que6's Official Comments were mainly focused on *hardware efficiency*; based on the comments, this can roughly be understood as the "amount" of hardware resources required per "unit problem size".
>
> It is debatable if basic metrics such as number of cores and transistors are meaningful for predicting the relative hardware efficiencies of processors with fundamentally different architectures (von Neumann versus neuromorphic) and characteristics [31], e.g., CPU cores focus on computation while neuromorphic cores integrate computation and memory.
>
> In any case, the veracity of our **claimed contribution** does not strongly depend on the hardware efficiency of Loihi 2 relative to other neuromorphic implementations or CPU. Specifically,
> - Due to our slack-free formulation, SF-HMVC will provably consume less hardware resources than QUBO-SNN on a neuromorphic computer, be it Loihi or other neuromorphic implementations that have higher hardware efficiency than Loihi.
> - Even if neuromorphic computing is fundamentally less hardware-efficient than the CPU, our results present clear evidence of the superior *energy efficiency* of SF-HMVC compared to the CPU solutions, which is a major benefit for applications where energy efficiency is paramount.
>
> Responding to Que6's questions:
> - An SNN allocates one neuron to handle a specific variable in the input problem. Thus, the maximum problem size that an SNN can solve is limited by the number of neurons that a neuromorphic computer can support. The limit on HMVC size that is solvable on Loihi 2 is thus due to limitations in the hardware implementation, and not due to a fundamental weakness of neuromorphic computing.
> - SF-HMVC is an SNN that we designed to operate on Loihi 2. We have added results that show that the simulation of SF-HMVC on CPU consumed several orders of magnitude more energy than SF-HMVC on Loihi 2. This at least shows that SF-HMVC could benefit from intrinsic neuromorphic processing in a way that its direct CPU simulation could not.
> - Note that there are CPU algorithms for solving HMVC that does not require QUBO reformulation and/or additional slack variables. The baseline method ILP-CPU which solves HMVC as integer linear programming using the Gurobi software is one such method (see Sec. 5.1 **Competitors**). ILP-CPU was capable of solving all the HMVC instances in our experiments, thus, arguably ILP-CPU is at least as "hardware-efficient" on the CPU as SF-HMVC is on Loihi 2. However, our results in Tables 3 and 4 show that ILP-CPU was *less energy-efficient* than SF-HMVC-Loihi.
>
>    It is possible that there exists a CPU analog of SF-HMVC that is more hardware-efficient on CPU than SF-HMVC is on Loihi 2, however, without providing sufficient details (e.g., pseudocode), it is not possible to prove or disprove the reviewer's claim/suspicion. More importantly, it is unknown if the CPU analog will be more *energy-efficient* than SF-HMVC-Loihi.

---

> > ### Comment · Reviewer_Que6 · 2024-08-13
> >
> > I would like to thank the authors for their responses. My concerns and lack of clarity have been largely addressed, and I found the paper interesting and potentially impactful. I am raising my score.

---

### Author Rebuttal · Authors · 2024-08-05

We thank the AC for handling our submission and the reviewers for their insightful comments.

K44F, cGCZ and 9cvo thought that the paper was technically solid. Que6 also did not report any technical flaws.

Que6's concerns were mainly on relevance and significance: note that our focus on spiking neural networks (SNN) for optimization is closely aligned with the Primary Area of "Optimization (convex and non-convex, discrete, stochastic, robust)" in NeurIPS 2024. Moreover, hypergraph minimum vertex cover (HMVC) is a fundamental combinatorial problem with many related formulations and practical applications. Thus, an effective SNN algorithm for HMVC will have major impact.

Also, we believe Que6 misread the results; the data in Tables 2 and 4 clearly show that the SNNs executed on Loihi (`QUBO-SNN` and `SF-HMVC-SNN`) consistently consumed much less energy than the CPU solutions (`ILP-CPU` and `QUBO-CPU`).

Based on a comment by K44F and 9cvo, we have added scalability analysis of the proposed SF-HMVC. Briefly, SF-HMVC scales linearly whereas the baseline QUBO-based SNN scales quadratically with the HMVC problem size. The analysis was empirically validated in the experiments, where `QUBO-SNN` was infeasible on Loihi 2 for the HMVC instances, whereas `SF-HMVC-SNN` could solve the same instances on Loihi 2 effectively (see Tables 3 and 4). In future iterations of neuromorphic hardware, `SF-HMVC-SNN` will retain its computational advantages over `QUBO-SNN`, due to our slack-free formulation for HMVC and handcrafted SNN design (Sec. 4).

More broadly, recent developments on neuromorphic technology (e.g., [1]) indicate the near-term availability of large-scale neuromorphic processors and/or clusters, hence, the current capacity limitation on Loihi 2 should not deter the NeurIPS community from researching SNN algorithms.

We also added the following results in the attached PDF:
- [Tables R1 and R2] Following a suggestion by Que6, to circumvent the inability to change the random seed on Loihi 2 and obtain statistics on the solution quality, we simulated the SNNs on the CPU via the Lava framework that allows changing the random seed (see `QUBO-Lava` and `SF-HMVC-Lava` in the uploaded PDF). Note that `QUBO-Lava` and `SF-HMVC-Lava` are not *instrinsic* CPU analogs of the SNN algorithms, hence, their performance (solution quality, energy consumption, runtime) are not reflective of what is achievable on the neuromorphic hardware.
- [Figure R3] Following a suggestion by cGCZ, we conducted ablation studies for the hyperparameters of `SF-HMVC-Loihi` on MVC and HMVC instances. The results show that there are hyperparameter configurations where our method consistently yielded high-quality solutions; see our response to cGCZ for more details of this experiment.

For more details of the above, please see our individual responses to the reviewers.

**Summary**

The reviewers found the paper technically solid and/or did not indicate technical flaws. We have addressed the main concerns on relevance, significance, scalability and hardware limitation, as well as obtained statistics on solution quality via SNN simulation on CPU. We thank the AC and reviewers again for their time.

---

### Decision · Program_Chairs · 2024-09-25

**Decision:**

Accept (poster)

**Comment:**

The paper presents a novel method for solving the Hypergraph Minimum Vertex Cover (HMVC) problem using Spiking Neural Networks on neuromorphic hardware. This method enables the neuromorphic system to solve small instances of the HMVC problem more effectively than previous methods, consuming less energy than traditional optimization techniques.

All reviewers agree that this paper presents a significant advance and vote for acceptance.